# GaN Laser Diode Technology for Visible-Light Communications

Stephen P. Najda [1,*], Piotr Perlin [1,2], Tadek Suski [2], Lucja Marona [1,2], Mike Leszczyński [1,2], Przemek Wisniewski [1,2], Szymon Stanczyk [2], Dario Schiavon [1], Thomas Slight [3], Malcolm A. Watson [4], Steffan Gwyn [5], Anthony E. Kelly [5] and Scott Watson [5]

1   TopGaN Ltd., Aleja Prymasa Tysiąclecia 98, 01-424 Warsaw, Poland; p.perlin@topganlasers.com (P.P.); l.marona@topganlsers.com (L.M.); m.leszczynski@topganlasers.com (M.L.); p.wisniwski@topganlasers.com (P.W.); d.schiavon@topganlasers.com (D.S.)

2   Institute of High Pressure Physics PAS, ul. Sokolowska 29/37, 01-142 Warsaw, Poland; tadek@unipress.waw.pl (T.S.); s.stanczyk@topganlasers.com (S.S.)

3   Sivers Photonics Ltd., 4 Stanley Boulevard, Hamilton International Technology Park, Blantyre, Glasgow G72 0BN, UK; thomas.slight@sivers-photonics.com

4   BAE Systems Advanced Technology Centre, FPC 267, Filton, Bristol BS34 7QW, UK; malcolm.watson@avoptics.com

5   James Watt School of Engineering, University of Glasgow, Glasgow G12 8LT, UK; s.gwyn.1@research.gla.ac.uk (S.G.); anthony.kelly@glasgow.ac.uk (A.E.K.); scott.watson@glasgow.ac.uk (S.W.)

*   Correspondence: s.najda@topganlasers.com

**Abstract:** Gallium nitride (GaN) laser diodes (LDs) are considered for visible light communications (VLC) in free space, underwater, and in plastic optical fibers (POFs). A review of recent results is presented, showing high-frequency operation of AlGaInN laser diodes with data transmission rates up to 2.5 Gbit/s in free space and underwater and high bandwidths of up to 1.38 GHz through 10 m of plastic optical fiber. Distributed feedback (DFB) GaN LDs are fabricated to achieve single-frequency operation. We report on single-wavelength emissions of GaN DFB LDs with a side-mode suppression ratio (SMSR) in excess of 35 dB.

**Keywords:** GaN laser; GaN DFB; GaN systems; underwater communications; plastic optical fiber communications

## 1. Introduction

Gallium nitride (GaN) light-emitting diodes (LEDs) have gained significant interest for use in visible-light communications (VLC) [1]. Nowadays, there is a push towards green, sustainable sources of communication, with VLC sources being able to provide both lighting and communication simultaneously. GaN LED-based visible-light communications typically provide low data rates [2]; however, recent work has shown that GaN micro-LEDs can have bandwidths of hundreds of MHz, allowing for Gbit/s data transmission [3]. By exploiting techniques such as wavelength division multiplexing, data rates of over 10 Gbit/s have been reported [4,5]. However, the performance of GaN LEDs is limited by moderate bandwidths, low power, and a rapidly divergent beam, resulting in short transmission distances and limiting potential system applications.

In comparison, GaN laser diodes (LDs) have shown much higher modulation frequencies, higher powers, and better beam quality, allowing for long-reach performance and paving the way for many new VLC applications to be realized. Previous work has shown that GHz data transmission is possible using directly modulated GaN LDs in free space, underwater, and through plastic optical fibers (POFs) [6,7]. In free space, data transmission rates exceeding 25 Gbit/s have been shown by using a 64-quadrature amplitude modulation discrete multi-tone (64-QAM DMT) system [8]. Extensive work has been carried out in underwater environments due to the low-loss transmission window through water in this part of the spectrum. A system was presented by Wu et al., where data rates of 12.4 Gbit/s

were achieved through water [9]. This technology could be used to provide communication between underwater autonomous vehicles for oil and gas exploration or could be used to assist docking stations or for seabed monitoring.

We also report here on recent developments in single-wavelength distributed feedback (DFB) GaN LDs, a topic garnering significant research interest for applications such as atomic cooling and optical communications [10,11]. Achieving narrow linewidths is a key priority to allow these devices to be used in quantum clocks, and specific wavelengths can be targeted for different applications. A number of groups around the world have focussed on ways to optimize the GaN DFB process [12–14].

## 2. Materials

GaN LDs used in this work were fabricated from AlGaInN epitaxy via metal organic chemical vapor deposition (MOCVD), consisting of (i) an 800 nm $Al_{0.08}Ga_{0.92}N$ lower cladding layer, (ii) 50 nm GaN lower waveguide layer, (iii) 50 nm $In_{0.02}Ga_{0.98}N$ injection layer, (iv) $In_xGa_{1-x}N/In_{0.02}Ga_{0.98}N$ quantum wells ×3 (3.5/9 Å) (the indium composition (x) and well thickness can be varied to change the emission wavelength), (v) 20 nm $Al_{0.2}Ga_{0.8}N$ electron blocking layer, (vi) 80 nm GaN waveguide, and (vii) 350 nm $Al_{0.08}Ga_{0.92}N$ upper cladding.

AlGaInN laser epitaxy structures were processed into 2 µm ridge waveguide laser diodes for single-transverse mode, with a cavity length of 700 µm.

## 3. Results

The optical power–current–voltage (LIV) characteristics for a 2 µm ridge waveguide LD structure are shown in Figure 1a and the single-transverse optical beam modes are shown for both the fast and slow axes in Figure 1b. An optical power in excess of 120 mW is shown with a threshold current of 50 mA and a threshold voltage of ~3.5 V. The spectral output at ~418.8 nm is shown in Figure 2, revealing a multi-longitudinal mode structure typical of a Fabry–Perot (FP) LD device.

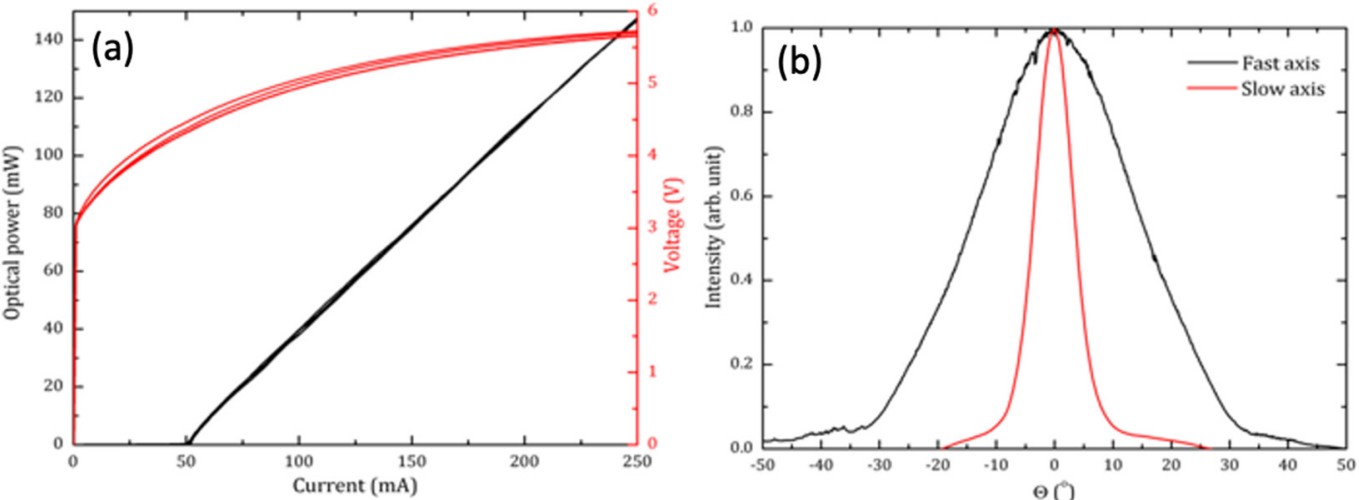

**Figure 1.** AlGaInN 418 nm laser diode characteristics: (**a**) LIV; (**b**) near-field (slow axis: red line; fast axis: black line).

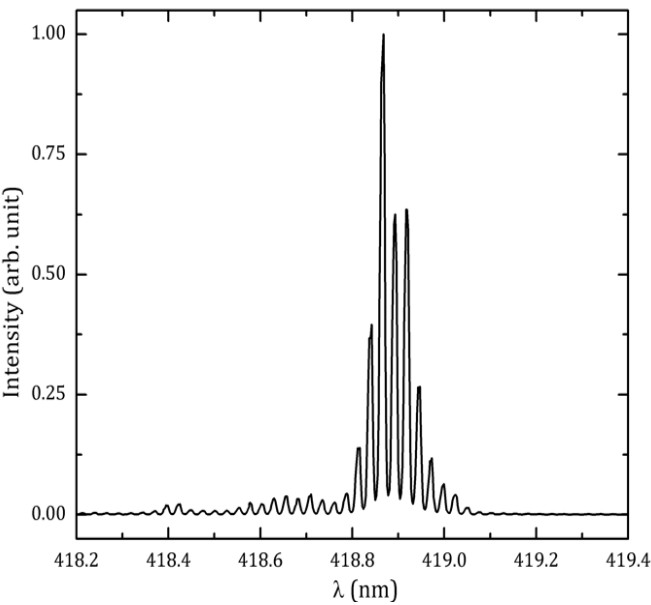

**Figure 2.** Spectral output of a 418 nm GaN LD.

### 3.1. GaN Laser Diode Free-Space Data Transmission

Data transmission measurements were conducted in free space using a 422 nm GaN LD. Figure 3 shows the eye diagrams measured using an Agilent 86105B digital sampling oscilloscope. High-speed data transmission (small signal modulation) at 1 Gbit/s was measured at a laser drive current of 115 mA (Figure 3a) and at 2.5 Gbit/s for 120 mA (Figure 3b), at which the best Q-factor margins were achieved.

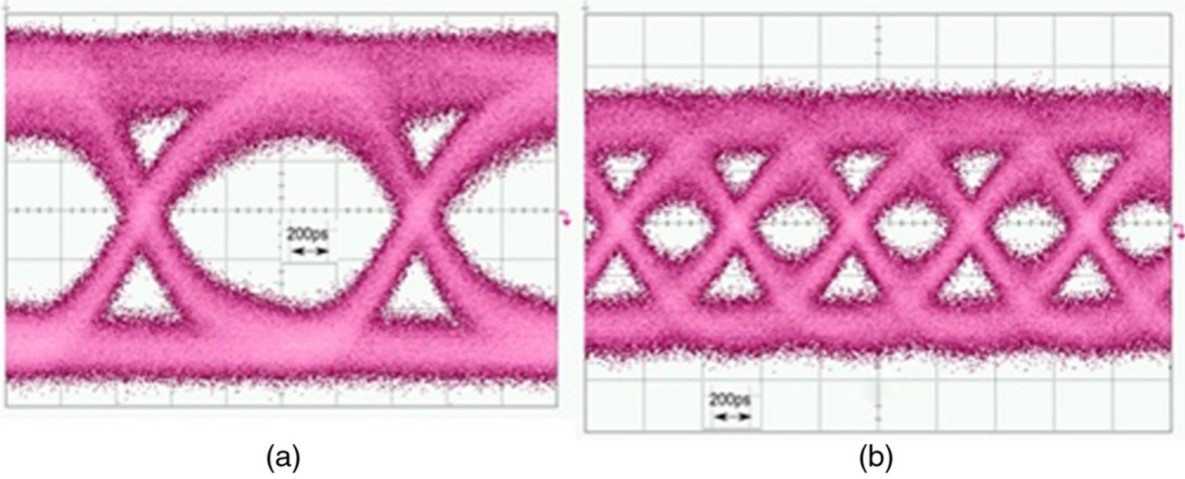

**Figure 3.** Eye diagrams at (**a**) 1 Gbit/s and (**b**) 2.5 Gbit/s at the photoreceiver output.

### 3.2. GaN Laser Diode Underwater Data Transmission

GaN LDs have great potential for use in underwater communications, as the attenuation coefficient through water is at its lowest in the blue-green part of the spectrum. To test the suitability of GaN LD technology for underwater communications, a GaN laser optical tracking system was constructed and submerged in a water tank (see Figure 4) [15]. Underwater experiments using a GaN LD have also been reported elsewhere [16–18].

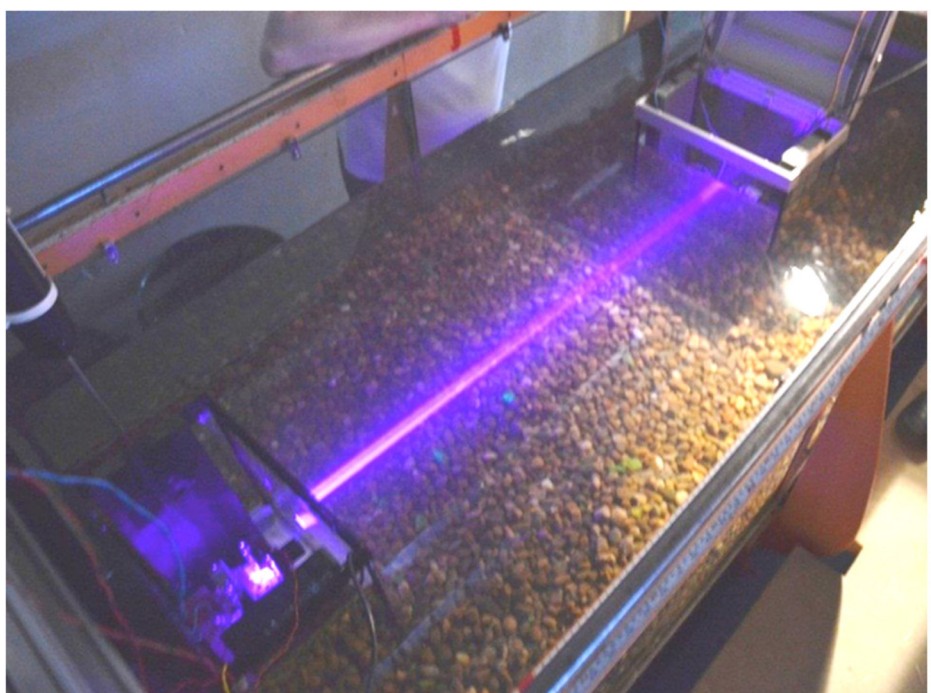

**Figure 4.** Collimated laser fired underwater from node to node in harbor-type water over a ~1 m distance.

The frequency response of the GaN LD was measured over an underwater path length of ~1 m, showing that GHz operation was possible. This resulted in high-speed data transmission at Gbit/s rates, demonstrating the suitability of the GaN system technology for underwater sensing and communications (see Figure 5).

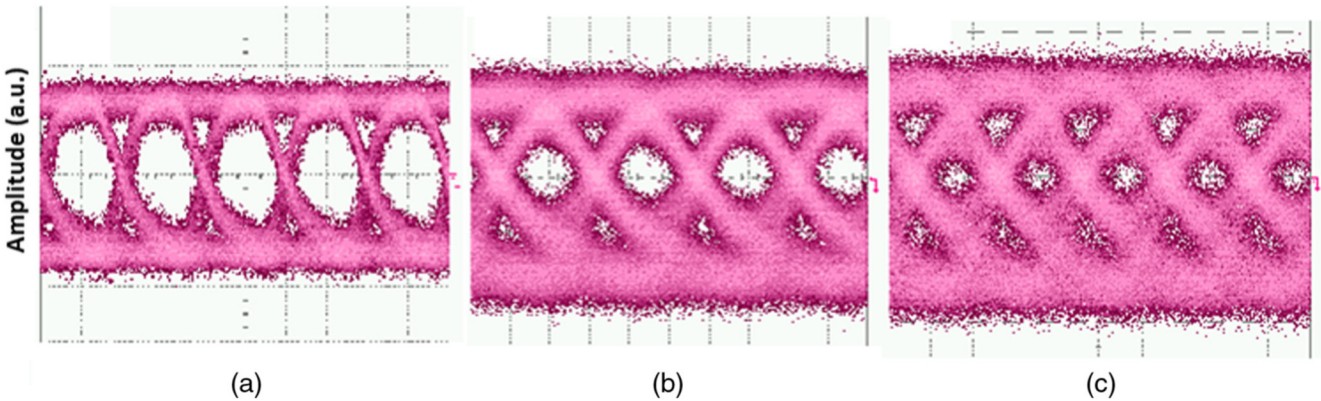

**Figure 5.** Eye diagrams showing data transmission for a signal transmitted through water at (**a**) 1 Gbit/s at 125 mA laser drive current, (**b**) 2 Gbit/s at 132 mA, and (**c**) 2.488 Gbit/s at 132 mA.

The relative openness of the "eyes" is a standard measure of the quality of data transmission. Through 'oceanic clear' water, a data rate of up to 2.488 Gbit/s underwater can be measured.

In order to test the performance of the GaN laser system in high-turbidity water conditions, Maalox was introduced to the water to mimic the volume scattering of seawater particles, which is representative of 'harbor' water.

It can be seen from Figure 6 that increasing the scattering solution results in an expected laser power transmission reduction through the water, reducing the optical intensity at the detector. However, this demonstrates that data transmission is possible, even in highly scattered water environments.

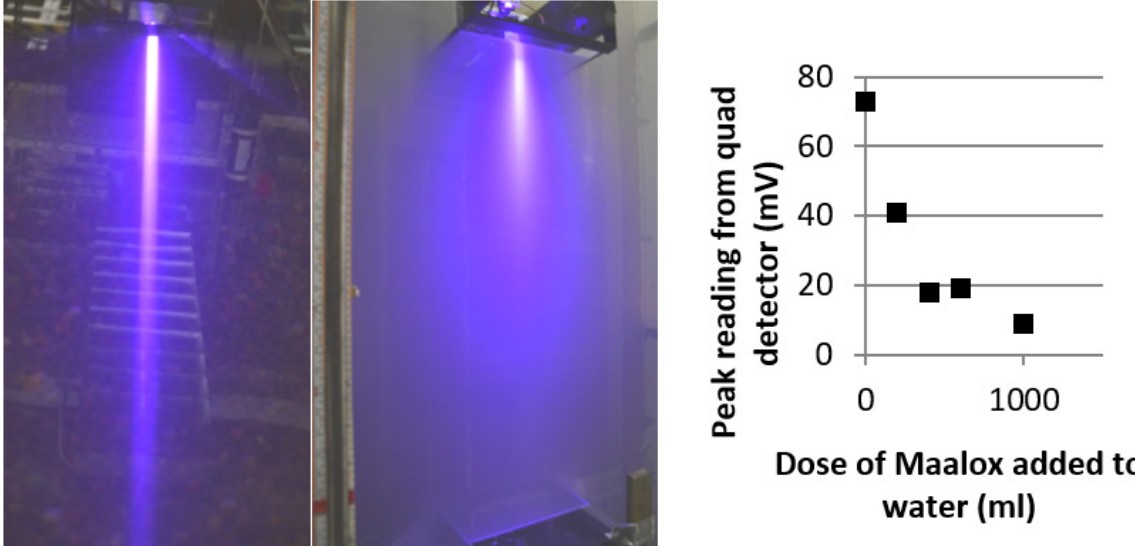

**Figure 6.** Laser transmission over 1 m of water with increasing scattering solution. (**Left**): Photo with a light concentration of Maalox added. (**Centre**): Very high concentration of Maalox added. (**Right**): Peak voltage reading from the quadrant detector representing received power over the underwater path length, for increasing Maalox concentration.

### 3.3. GaN Laser Diode Data Transmission in Plastic Optical Fibers (POFs)

Visible-light data transmission using GaN LEDs through POFs has been reported previously [19,20]. Here, we report on data transmission in POFs using a GaN LD.

A GaN LD emitting at 429 nm was used to conduct frequency response measurements through various lengths of 1-mm-diameter step index plastic optical fibers (SI-POFs). Fiber lengths of 1, 2.5, 5, and 10 m were tested to discover the relationship between the bandwidth and fiber length. This laser had a −3 dB bandwidth of 1.71 GHz in free space and error-free data transmission at 2.5 Gbit/s was achieved in a similar manner to that reported above. The maximum bandwidth values achieved for transmission through 1, 2.5, 5, and 10 m of fiber were 1.68, 1.63, 1.62, and 1.1 GHz, respectively, as seen in Figure 7a.

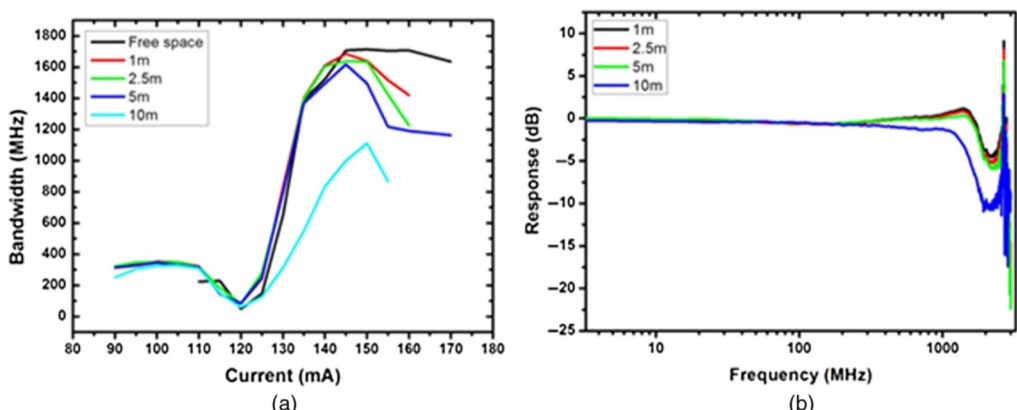

**Figure 7.** (**a**) Current versus bandwidth relationship for varying lengths of SI-POF and (**b**) fiber as a function of length.

The bandwidth of the fiber itself can be obtained by subtracting the free space response from that measured through the different lengths of fiber. This allows for dispersion analysis to be carried out. It should be noted that the LD was not optimized for high-frequency operation, nor was the laser optimized for fiber coupling so that further improvements could

be made. In Figure 7b, the bandwidth of the fiber measuring 10 m drops to 1.38 GHz due to modal dispersion, showing that Gbit/s transmission is still possible over these distances.

### 3.4. Visible DFB GaN LDs

Distributed feedback (DFB) lasers in the infrared range are successfully used in many telecommunication applications due to the narrow spectral linewidth and mode stability of the device. In comparison, DFB GaN LDs in the visible range are much more challenging to fabricate and have received less attention for communication applications. DFB GaN LDs can be fabricated using a buried or surface grating on a chip to select the longitudinal mode. However, both methods have their disadvantages—complex overgrowth steps are required for buried gratings, which have the potential to introduce epi-defects, while surface grating designs can compromise the quality of the *p*-type top contact and suffer from increased optical losses in unpumped grating regions. In this approach, gratings are etched onto the sidewalls of a ridge waveguide laser diode. The sidewall grating can be designed and implemented entirely post-growth once the emission wavelength is known. Additionally, the coupling coefficient is mainly determined by the planar layout of the grating rather than the etch depth, unlike surface or buried gratings, allowing more freedom in design [21,22]. A similar approach to fabricating DFB GaN LDs has been described elsewhere [12–14].

### 3.4.1. Design and Fabrication of Visible DFB GaN LDs

One of the key challenges in designing a GaN-based DFB is attaining the required Bragg wavelength in the etched grating. As the emission wavelength of these devices is ~400 nm, a first-order grating would require a feature size of ~40 nm, which is not currently feasible. To combat this, third-order gratings with complementary feature sizes of ~120 nm were used, and an 80% duty cycle was implemented, such that loss of coupling strength by using a lower-order grating was minimized, as seen in Figure 8a [21]. Then, to calculate the coupling coefficient $\kappa$ values of these devices, the effective indices of the wide and narrow sections of the sidewall gratings at 2.5 μm and 1.5 μm, respectively, using coupled mode theory [21] yield a coupling strength of $\kappa = 22$ cm$^{-1}$.

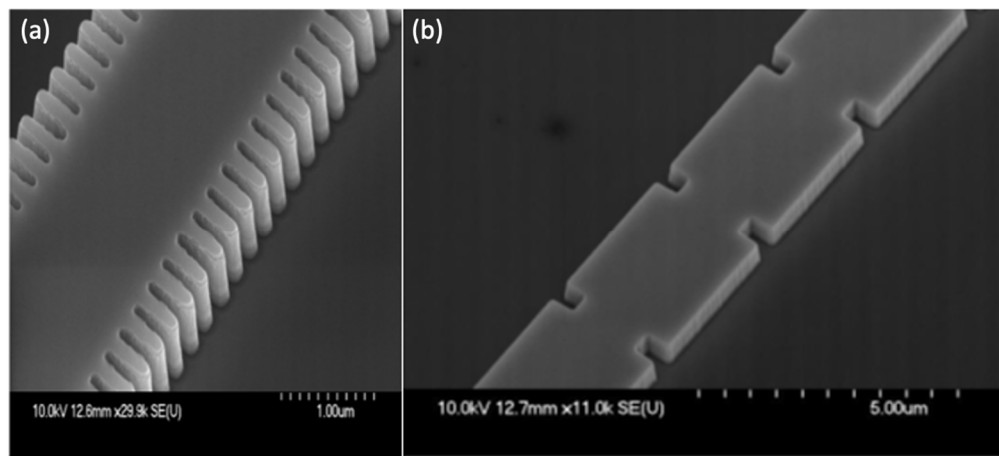

**Figure 8.** (**a**) SEM image of the as etched third-order DFB grating and (**b**) SEM image of the as-etched 39th DFB-order grating.

Another way to achieve single-mode emission is by utilizing a higher-order grating. The operation involves the use of a Fabry–Perot (FP) laser with weak optical feedback provided by a high-order grating that runs along the partial length of the waveguide. The grating feedback is sufficient to allow lasing in a single or narrow band of FP modes, which are close in wavelength to the Bragg wavelength, $\lambda_b$, with FP modes away from $\lambda_b$ experiencing an increased loss penalty. In order to ensure single-wavelength operation, the grating must have a reflection bandwidth that is on the same order as the free spectral range

(FSR) of the cavity (<0.1 nm). This design makes use of a 39th-order grating with 125 notch pairs along the ridge, which we estimate to have a bandwidth of ~0.11 nm (Figure 8b).

Devices were fabricated from AlGaInN laser epi-structures, consisting of three In-GaN quantum wells sandwiched between GaN barriers, GaN waveguide layers, and $Al_{0.06}Ga_{0.94}N$ cladding layers. The quantum wells were designed to emit around 410 nm.

To fabricate the gratings, electron beam lithography was used to define the ridge or grating pattern and ICP etching was carried out. An optimized ICP process with a $Cl_2/N_2$ chemistry gives a smooth and vertical etching profile, which is important to achieve optimal grating performance (see Figure 8). Before cleaving into individual chips, the sample was thinned and polished and the back metal was deposited. We fabricated 'standard' 2 μm ridge waveguide Fabry–Perot (with no grating) GaN LDs on the same wafer adjacent to the DFB GaN LDs for comparison.

### 3.4.2. 39th-Order DFB GaN LD Characterization

The LI characteristics of a 39th-order DFB GaN LD are compared to a standard FP GaN LD fabricated side-by-side in Figure 9. The standard FP GaN LD has a threshold current of 70 mA and slope efficiency of 0.6 W/A, compared to a DFB GaN LD of 130 mA and 0.27 W/A. An increase in threshold and decrease in slope efficiency by almost a factor of two can be observed in the LI characteristics of the 39th-order DFB compared to the FP GaN LD due to the increased scattering of the grating.

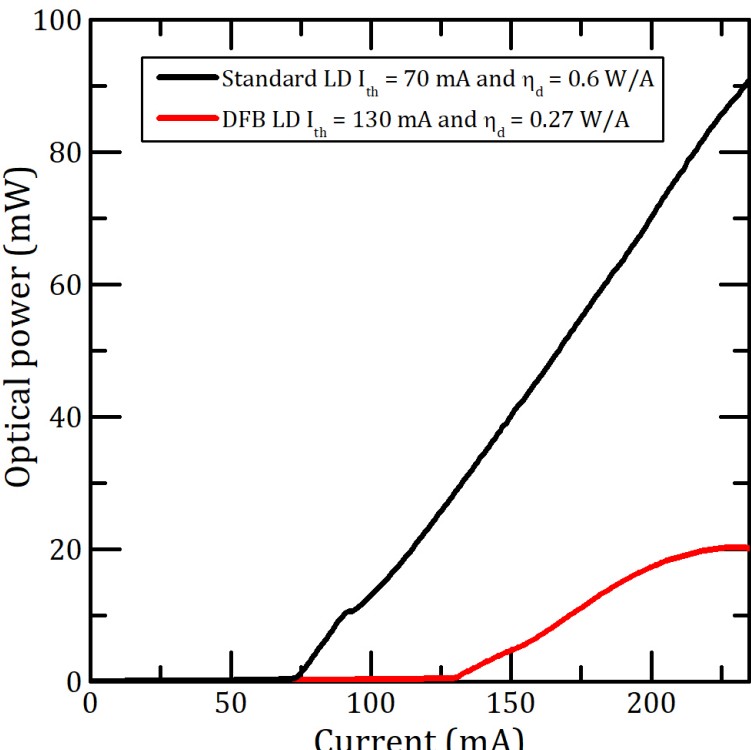

**Figure 9.** Comparison of the LI characteristics between a standard FP LD (black) and a 39th-order DFB GaN LD (red).

Figure 10 shows the spectral evolution of the 39th-order DFB GaN LD as a function of the drive current. From 150 mA to 225 mA, single longitudinal mode selectivity is observed in this device.

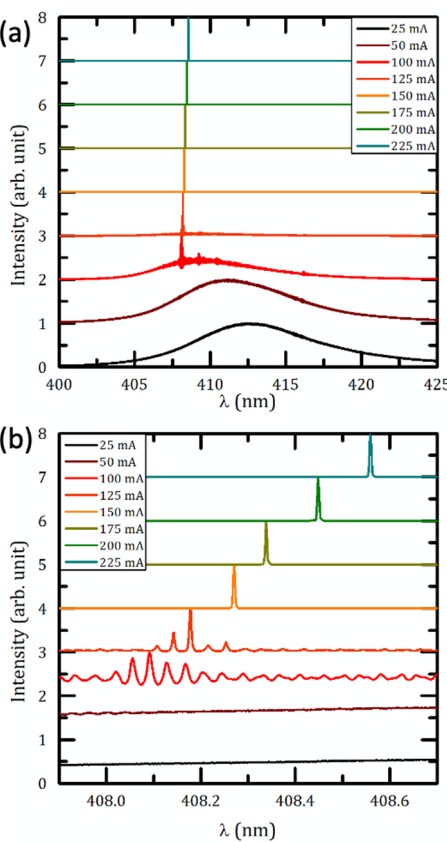

**Figure 10.** (**a**) Emission spectra for a 39th-order DFB GaN LD at different driving currents shown over a large wavelength range and (**b**) a small wavelength range.

### 3.4.3. Third-Order DFB GaN Laser Diode Characterization

Fabry-Perot GaN LDs were fabricated adjacent to third-order DFB GaN LDs and the LIV characteristics are compared in Figure 11. At a threshold current of 22–27 mA, output powers of 80–90 mW at a turn-on voltage of ~3.5 V were observed for the FP GaN LDs (D67, D68). The third-order DFB GaN LD (D70) characteristics show a higher threshold, lower power, and increased turn-on voltage compared to their FP counterparts due to the increased scattering from the etched grating.

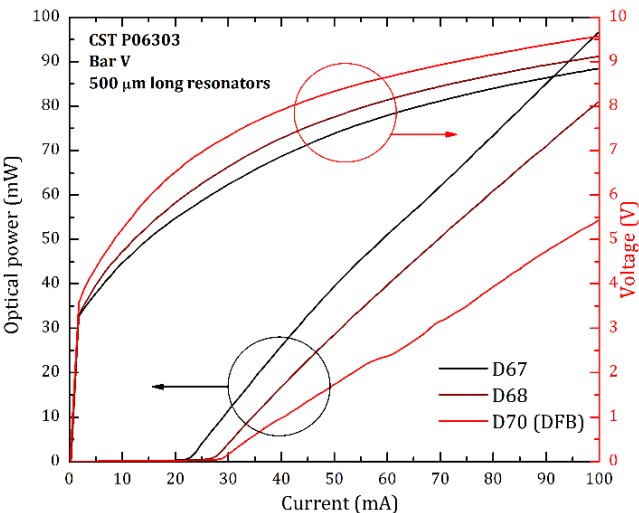

**Figure 11.** LI characteristics of a FP GaN LD versus a third-order DFB GaN LD.

The spectral output of a third-order DFB GaN LD as a function of the drive current is shown in Figure 12. From 45 to 70 mA, single-mode operation can be observed. A side-mode suppression ratio (SMSR) in excess of 35 dB was achieved, as seen in Figure 13.

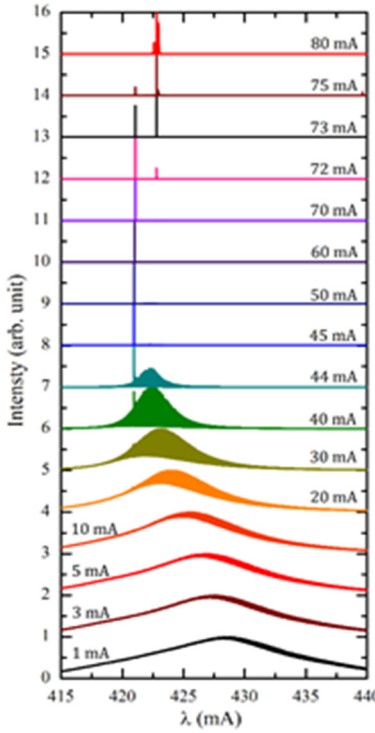

**Figure 12.** Spectra of a third-order DFB GaN LD vs. drive current.

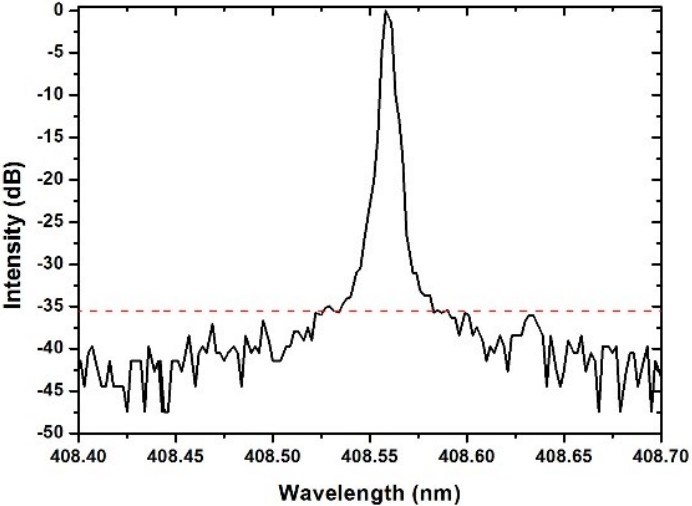

**Figure 13.** Spectral performance of a third-order DFB GaN LD exhibiting a SMSR exceeding 35 dB.

The frequency response of the third-order DFB GaN LD is shown in Figure 14. A maximum bandwidth of 2.3 GHz was achieved at 80 mA. The bandwidth began to decrease after this point, reaching 1.2 GHz at 140 mA. The bandwidth was limited by the parasitic capacitance within the package of the device, which was not optimized for high-frequency operation, so there is scope to improve this further.

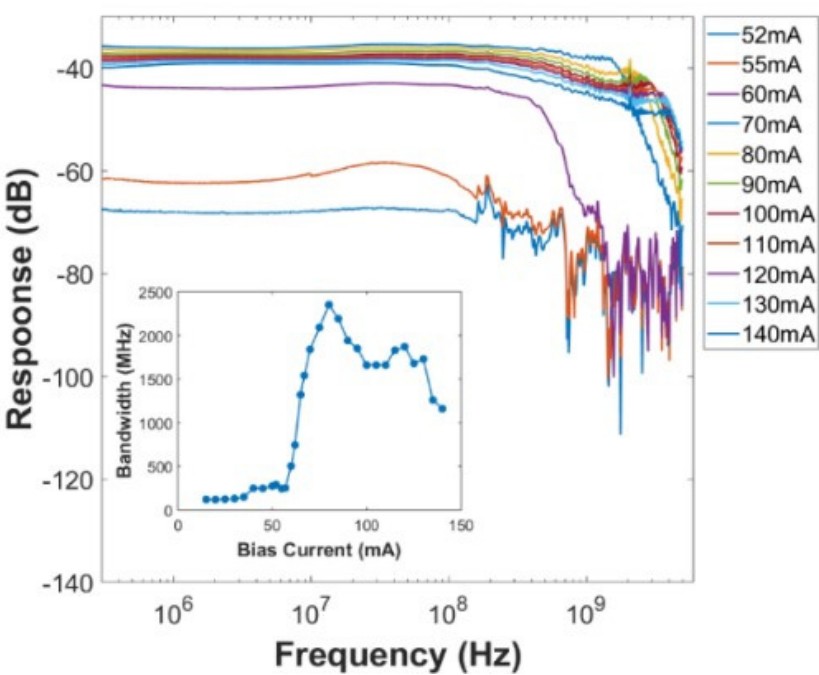

**Figure 14.** Frequency response of the third-order DFB GaN LD as a function of the drive current and inset of the −3 dB optical bandwidth as a function of the bias current.

Eye diagrams were taken for this device at varying data transmission rates. Error-free transmission was achieved up to 3 Gbit/s and transmission at 2 Gbit/s is shown in Figure 15.

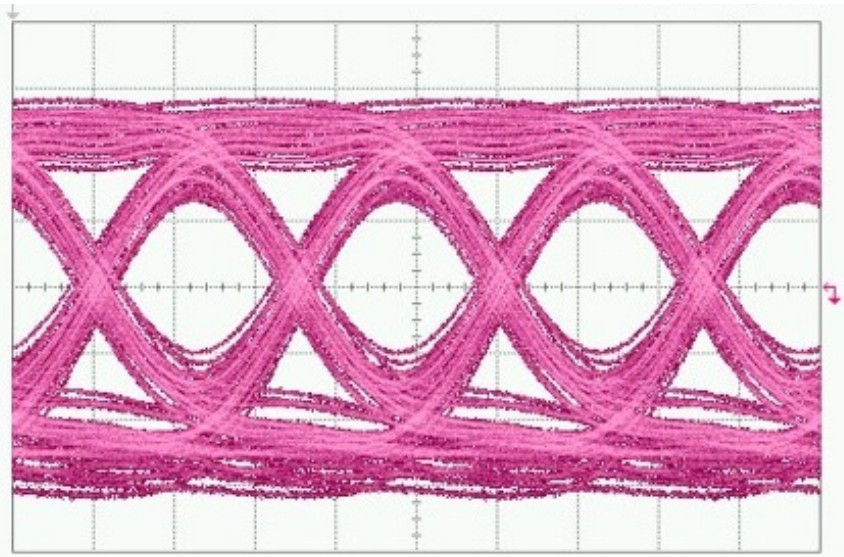

**Figure 15.** Eye diagram from the third-order GaN DFB LD at 2 Gbits/s showing error-free communication.

## 4. Conclusions

GaN LDs have great potential for many visible-light communication applications, including free-space, underwater, and POF applications. Data rates of up to 2.5 Gbit/s have been reported here using a blue GaN LD in free space, underwater, and with POFs. Furthermore, the development of DFB GaN LDs for single-mode operation have been shown using a third-order strongly coupled grating and a weakly coupled 39th-order grating, demonstrating the potential for DFB GaN LDs for telecommunication purposes.

**Author Contributions:** Conceptualization, A.E.K., S.W.; methodology, A.E.K., S.W., S.G., S.P.N., P.P., T.S. (Tadek Suski), L.M., M.L., P.W., S.S., D.S., T.S. (Thomas Slight), M.A.W.; software, A.E.K., S.G., S.W.; validation, S.W.; formal analysis, S.W., A.E.K.; investigation, S.W.; resources, S.P.N., M.A.W., A.E.K.; data curation, S.W.; writing—original draft preparation, S.P.N. and S.W.; writing—review and editing, S.P.N. and S.W.; visualization, S.W.; supervision, A.E.K.; project administration, A.E.K.; funding acquisition, A.E.K. and S.P.N. All authors have read and agreed to the published version of the manuscript.

**Funding:** This research received no external funding.

**Acknowledgments:** This research was supported by the European Union with Grant No. E!113612 and the National Center for Research and Development within the project E!113612/NCBiR/2021.

**Conflicts of Interest:** The authors declare no conflict of interest.

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
