# Peer review of "GaN Laser Diode Technology for Visible-Light Communications"

_electronics, doi:10.3390/electronics11091430_

Round 1

Reviewer 1 Report

In this paper, the authors overview recent results of performance of GaN laser diodes and their applications for visible light communications. They cover both device technologies and performances for a practical use, and provide a useful information to the readers in this research field. As a review article, the paper is encouraged to present more detailed descriptions at each technical discussion. Please find the following comments.

  1. P1, L23: The authors are encouraged to describe more comprehensive research background such as a history of this research field, other competitive research, social demand and etc.
  2. P7, L176: A 39th order DFB laser an increase in threshold and decrease of slope efficiency. The authors are encouraged to discuss about the reasons of the degradation and prospects of improvements.
  3. P8, L194: A 3rd order DFB laser a higher threshold, lower power and increased turn-on voltage compared to the FP laser. The authors are encouraged to discuss about the reasons of the degradation and prospects of improvements. Also, why does it shows a lower threshold than the 39th order DFB laser?
  4. P2, L209: Without the parasitic capacitance how much would the bandwidth be extended? And, how much is the capacitance at the present package?
  5. P4, L102: the relationship between “5 parts in 10^4 part water” and the amount of dose in Fig. 6 is not clear. Please show the relation.

  1. P2, L43: Is “AlGaInN” the same meaning as “GaN”? If so, these expressions should be unified in this article.
  2. P2, L44: “0.8 um” should read “800 nm”.
  3. P2, L55: “(FP)” should be added after “Fabry-Perot”.
  4. P3, L77: “GaN LD’s” should read “GaN LD”. (“‘s” is no need.)
  5. P3, L78: “GaN laser diode” should read “GaN LD”.
  6. P4, L87: “Gbits/s” should read “Gbit/s”. (“s” is no need.)
  7. P4, L105: “Figure 6” should read “Fig. 6”.
  8. P4, L106: “transfer” should read “transmission”.
  9. P5, L131: “GaN LD’s” should read “GaN LD”. (“‘s” is no need.)
  10. P6, L144: “GaN LD’s” should read “GaN LD”. (“‘s” is no need.)
  11. P6, L149: There are no explanation about Fig. 8(a) in the text. “Fig. 8(a)” may be better to added after “minimized [11]”.
  12. P6, L155: “(lambda b)” should be added after “the Bragg wavelength”.
  13. P6, L158: “Fig. 8” should read “Fig. 8(b).
  14. P6, L159: This paragraph contains only two sentences. It should be merged with the next paragraph.
  15. P6, L166: “GaN LD’s” should read “GaN LD”. (“‘s” is no need.)
  16. P7, L173: “GaN laser diode” should read “GaN LD”.
  17. P7, Fig. 10: There seems to be lasing at 125mA. However, there seems to show no power at 125mA. Please explain the inconsistence.
  18. Please explain about “D67, D68, D70 (DFB)” in Fig. 11.
  19. P9, L205: What is “drive current”? Isn’t it small signal responses?
  20. P10, L210: This paragraph contains only two sentences. It should be merged with the previous paragraph.

I will appreciate it if you accept all suggested corrections, and resubmit a revised manuscript.

Reviewer 2 Report

This review summarizes a few research works on GaN laser technology used for visible light communication. The topic of this review is certainly interesting, however, the content can be further enriched. For example, in the conclusion, the sentence “we report up to...”  and "we have developed" makes this review like a summary of their own research in an article paper. This review only included 17 references, this is much less than expected in a good review paper. Other minor concerns are:

Line 42-50: the authors could consider adding a figure illustrating the configuration of the devices, which will make the paper more readable.

I suggest the authors reorganize and merge some figures, instead of showing one single figure each time.

Round 2

Reviewer 1 Report

Thank you for your revised manuscript. I think it is not your final revised version. It is because many points are not revised although you agreed with my comments about them. For example, comments #4, 7, 8, 9, 10, 11, 13, and so on. 

Author Response

Changes made as requested see attached updated manuscript and referee 1 comments with replies

Reviewer 2 Report

Thank you for addressing the comments.

Author Response

Please see attached document for comments

Round 3

Reviewer 1 Report

The manuscript is revised according to the comments. By the way, I still have the following comments.

Figure 7(a): There is no explanation about this figure. Explanation about Fig. 7(a) should be added.

My previous comments: “GaN laser diode” should read “GaN LD”.

The authors’ response: Ok – a mix of ‘LD’ and ‘laser diode’ have been used at appropriate points.

However, I do not think they are used at appropriate points, for example, at section 3.4.2.

Author Response

I have submitted a reply to Reviewer 1 & updated paper

This manuscript is a resubmission of an earlier submission. The following is a list of the peer review reports and author responses from that submission.